# The Canine Pancreatic Extracellular Matrix in Diabetes Mellitus and Pancreatitis: Its Essential Role and Therapeutic Perspective

**DOI:** 10.3390/ani13040684

**Published:** 2023-02-15

**Authors:** Bruna Tássia dos Santos Pantoja, Rafael Cardoso Carvalho, Maria Angelica Miglino, Ana Claudia Oliveira Carreira

**Affiliations:** 1Department of Surgery, School of Veterinary Medicine and Animal Science, University of Sao Paulo, Sao Paulo 05508-270, SP, Brazil; 2Department of Animal Science, Center for Agricultural and Environmental Sciences, Federal University of Maranhao, Chapadinha 65500-000, MA, Brazil; 3Center for Natural and Human Sciences, Federal University of ABC, Santo Andre 09280-550, SP, Brazil

**Keywords:** decellularization, ECM remodelling, endocrine and exocrine diseases, pancreas, scaffolds, tissue bioengineering

## Abstract

**Simple Summary:**

The dog is considered an animal model for the study of several diseases that occur in humans since they present similar phenotypic development. Among them, we can highlight diabetes mellitus and pancreatitis, which are diseases that affect the endocrine and exocrine portion of the pancreas, respectively, and showing high prevalence, social cost, mortality, and morbidity in companion animals. This work aimed to highlight the importance of using the dog as a model for the study of changes in the pancreatic extracellular matrix when affected by diabetes mellitus and pancreatitis. The extracellular matrix performs several functions, such as physical support and regulation of cellular processes, being composed mainly of proteins, glycoproteins, glycosaminoglycans, and proteoglycans. It is noteworthy that there are no studies characterizing the healthy and diseased canine pancreatic extracellular matrix, as well as studies related to the matrix components involved in the progression of these diseases. It is known that most pathological pancreatic conditions induce extracellular matrix changes through a remodelling process, which has to be thoroughly studied to fully understand the pathogenesis of any pancreatic disease.

**Abstract:**

Diabetes mellitus and pancreatitis are common pancreatic diseases in dogs, affecting the endocrine and exocrine portions of the organ. Dogs have a significant role in the history of research related to genetic diseases, being considered potential models for the study of human diseases. This review discusses the importance of using the extracellular matrix of the canine pancreas as a model for the study of diabetes mellitus and pancreatitis, in addition to focusing on the importance of using extracellular matrix in new regenerative techniques, such as decellularization and recellularization. Unlike humans, rabbits, mice, and pigs, there are no reports in the literature characterizing the healthy pancreatic extracellular matrix in dogs, in addition to the absence of studies related to matrix components that are involved in triggering diabetes melittus and pancreatitis. The extracellular matrix plays the role of physical support for the cells and allows the regulation of various cellular processes. In this context, it has already been demonstrated that physiologic and pathologic pancreatic changes lead to ECM remodeling, highlighting the importance of an in-depth study of the changes associated with pancreatic diseases.

## 1. Introduction

The canine species has a significant role in the history of diabetes mellitus (DM) research since it was the first species to experimentally induce the disease. According to the state of pet health by Banfield’s 2021 [1] annual report, the diagnosis of being overweight or obese has increased significantly in the last 9 years, making dogs 3.7 times more likely to develop endocrine disorders such as diabetes mellitus. DM is predominantly of chronic autoimmune origin in dogs and can be fatal if not diagnosed and properly treated since the animals are completely dependent on insulin therapy to survive [2].

As in the study of DM, dogs have also been used as an animal model for the study of pancreatitis. Dogs tend to develop the disease spontaneously, being phenotypically similar to the development in humans, both related to genetic predisposition and environmental change [3]. Unlike diabetes mellitus, which is an endocrine disease, pancreatitis affects the exocrine portion of the pancreas and is characterized by inflammation in the exocrine pancreas, and in dogs, it can be classified as acute or chronic.

According to the latest update of the Online Mendelian Inheritance in Animals database [4], there are 546 genetic diseases in dogs that can be used as potential models for human diseases, among which we can highlight diabetes (mellitus and insipidus) and pancreatitis, emphasizing the importance of exploring the use of the dog as an animal model.

It is known that pathological conditions cause alterations in cellular function in addition to dysregulation of extracellular matrix (ECM) components. The ECM is a network of macromolecules that provides physical support and regulates a series of cellular processes such as migration, differentiation, survival, homeostasis, and morphogenesis [5,6,7]. In addition, it has a fundamental role in the remodelling of the response to physiological changes, acting in the structural modulation and the matrix properties during a pathological situation [8].

In this study, with the knowledge of the severity of these diseases and that dogs can be used as an experimental model, we seek to highlight the importance of using the extracellular matrix of the canine pancreas as a model for the study of diabetes mellitus and pancreatitis in humans.

## 2. Canine Pancreas

In dogs, the pancreas is V-shaped and is located along with the duodenum [9]; it is composed of a right lobe, a left lobe, and a central portion called the body of the pancreas that joins the two lobes [10]. Anatomically, the left lobe is positioned caudomedially, crossing the median plane behind the stomach and ending in contact with the left kidney. While the right lobe lies caudodorsally, following the dorsal surface of the duodenum, it is dorsally related to the visceral surface of the liver and the ventral surface of the kidney [11].

The pancreas is derived from the foregut and develops from the dorsal and ventral endodermal buds. At the time of the first rotation of the stomach along the longitudinal axis, the ventral pancreatic bud is displaced dorsally, staying close to the dorsal bud, after which the two brutes merge to form a single organ [12]. The dorsal bud gives rise to the largest portion of the pancreas including the right and left lobes as well as the body of the pancreas, while the ventral bud gives rise to a portion of the body of the pancreas.

The dorsal bud grows in the mesoduodenum and is arborized as endodermal cell cords, later forming a lumen [12]. In dogs, the ventral bud gives rise to the pancreatic duct that opens into the greater duodenal papilla along with or close to the bile duct [11]. Meanwhile, the main duct coming from the dorsal bud is the accessory pancreatic duct that opens into the minor duodenal papilla on the opposite side of the intestine, which is considered the major excretory pancreatic duct. The endoderm of the shoots gives rise to the exocrine acini and the endocrine islets of Langerhans [12]. The latter are formed by groups of endodermal cells that gradually lose their connections with the exocrine system.

The pancreas contains endocrine and exocrine elements [10], the exocrine portion is responsible for the production of digestive enzymes or their precursors that act in the degradation of fats, carbohydrates, and proteins [13]. The endocrine portion is composed of the islets of Langerhans, which are a cluster of cells distributed among the pancreatic acini, which are responsible for the production of hormones [11] such as insulin, glucagon, somatostatin, and pancreatic polypeptide [14].

The exocrine portion of the pancreas is similar to the parotid gland, what differentiates them histologically is the absence of striated ducts and the presence of the islets of Langerhans in the pancreas [15]. The pancreatic acini are surrounded by a basal lamina supported by a delicate sheath of reticular fibers and are made up of several serous cells that surround a lumen. This portion also has a wide network of capillaries essential for the secret process.

The exocrine component acts in the production of digestive enzymes that degrade proteins (trypsin, chymotrypsin, elastase, aminopeptidase, and carboxypeptidase), carbohydrates (amylase), and fats (lipase). In addition to enzymes, the exocrine portion secretes significant amounts of bicarbonate into the duodenum to help neutralize the acidic contents of the stomach and to maintain adequate pH in the duodenum for enzymes to act [16]. Therefore, pancreatic exocrine secretion is regulated by two hormones (secretin, and cholecystokinin), which are produced by enteroendocrine cells present in the intestinal mucosa [15].

The endocrine portion is constituted by the pancreatic or Langerhans islets, being composed of polygonal cells arranged in cords, around an abundant network of blood capillaries with fenestrated endothelial cells [15]. In addition, pancreatic islets have a thin layer of connective tissue that separates them from the rest of the pancreatic tissue. The blood with the secreted hormones passes through the capillary network of the acini in the exocrine portion before leaving the pancreas, allowing the hormones to act in the regulation of the exocrine pancreas [17].

Four cell types were identified in the islets of Langerhans: alpha cells (20−30%), which secrete glucagon; beta cells (60−80%), which secrete insulin; delta cells (5−10%), which secrete somatostatin; and F or PP cells (rare or absent) that secrete pancreatic polypeptide [17,18]. In dogs, beta cells are normally distributed peripherally and their proportion tends to increase with age, while alpha cells are located more centrally, and delta cells are randomly arranged and tend to decrease with age [17,19].

Alpha cells (α) are responsible for the secretion of glucagon. This hormone helps to keep blood glucose constant, that is, when the blood glucose level drops, glucagon secretion occurs to restore the level of glucose in the circulation [14]. Glucagon, produced by alpha cells in the pancreas, antagonizes insulin by mobilizing glucose from the liver through gluconeogenesis and glycogenolysis [16]. Beta (β) cells produce insulin (Figure 1), which is the main anabolic hormone of mammals and has two important functions: to stimulate carbohydrate and lipid metabolism by inducing cellular enzymes and to transport glucose across the plasma membranes of insulin-sensitive cells [20].

Delta cells (δ or D) produce somatostatin which is responsible for inhibiting insulin, glucagon, and growth hormone (GH) secretion and is necessary to decrease the activity of the gastrointestinal tract [16]. The F or PP (pancreatic polypeptide cells) cells secrete the pancreatic polypeptide, which acts in gastric secretion and emptying, in addition to the secretion of pancreatic enzymes [21]. Therefore, any dysfunction involving one of these cell lines can result in an excess or deficiency of the respective hormone in the circulation.

## 3. Diabetes Mellitus

Diabetes mellitus is characterized by a disorder that results in the inability of the pancreatic islets to secrete insulin and/or the deficient action of insulin in the tissues, being classified as insulin-dependent or type 1 diabetes, or non-insulin-dependent or type 2 diabetes. However, the classification that considers insulin dependence is not helpful in the canine patient, as all diabetic dogs become insulin-dependent, with few exceptions [22]. In both situations, a condition of insulin activity deficiency is established and glucose uptake is severely compromised [23].

In both dogs and humans, diabetes is a multifactorial disease involving genetic [24,25] and environmental factors [22,26,27]. Although the most frequent form in dogs is caused by autoimmunity, that is, by immune-mediated, rapid, and progressive destruction of beta cells [23].

Recently, the European Society of Veterinary Endocrinology (ESVE) established the ALIVE Project (agreeing language in veterinary endocrinology) to clarify common terminology and definitions used in veterinary endocrine diseases. Among the diseases addressed are DM, in which there are two etiological classifications for DM in dogs: DM with insulin deficiency (reduced insulin secretion and defective insulin production) and insulin-resistant DM (endocrine influence, obesity, drugs, medications inflammation, and disorders of the receptor and intracellular signaling) [28].

Insulin-deficient or type 1 DM is characterized by the destruction of β cells, with progressive and eventually complete loss of insulin secretion. It is considered the most common form in dogs. The cause of diabetes mellitus in dogs is poorly characterized, however, initial factors such as genetic predisposition, infection, disease- and insulin-antagonistic drugs, obesity, immune-mediated mechanisms, and pancreatitis have been identified [2].

In dogs, these factors result in the loss of beta cells and hypoinsulinemia, impairing glucose transport to most cells and thus accelerating the mechanisms of gluconeogenesis and hepatic glycogenolysis. Ketoacidosis develops as the production of ketone bodies increases to compensate for the underutilization of blood glucose. The loss of β-cell function is irreversible in dogs with type 1 DM, requiring lifelong insulin therapy to maintain glycemic control.

Insulin-resistant or type 2 DM is characterized by insulin resistance and/or dysfunctional beta cells. In type 2 DM, there is a high basal concentration of glucose in the blood and a basal concentration of insulin that can be high, low, or normal, however, the endogenous insulin release is insufficient to overcome the insulin resistance in the tissues [20]. This type of diabetes is not clinically recognized in dogs since this species is not susceptible to beta cell failure when there is a slight and persistent increase in blood glucose levels (18 mg/dL) [29].

Most dogs diagnosed with diabetes are 4 to 14 years of age, with the peak prevalence being between 7 and 9 years of age [2]. The most common clinical signs are polyuria/polydipsia, weight loss, and polyphagia, with wasting of the dorsal muscles, oily skin and cataracts [30]. Polyuria and polydipsia develop when the blood glucose concentration exceeds the renal threshold for absorption (12 to 14 mmol/L) [31]. Polyphagia and weight loss occur due to hyperglycemia and glucosuria, that is, when there is the mobilization of fat and protein reserves for gluconeogenesis, thus causing weight loss, hepatomegaly, and sarcopenia [31].

Unspayed females are three times more affected than males [32], once they go through the hormonal changes of the diestrus in which there is a predominance of progesterone exerting an antagonistic effect on insulin, as well as the GH released by the canine mammary glands under the influence of progesterone [33]. However, when castrated, there is a reduction in the number of diabetes cases as a result of insulin antagonism during the diestrus phase [31]. In this case, the progesterone source is removed and the plasma growth hormone concentration is reduced, causing the insulin antagonism to resolve [2].

The Poodle, Pinscher, Miniature Schnauzer, Dachshund, and Beagle breeds are the breeds with the highest incidence of DM [30]. Type 1 DM has a strong genetic association with genes encoding major histocompatibility complex (MHC) class II proteins, involved in antigen presentation to the immune system [34]. Genetic studies highlight that some MHC alleles are associated with an increased risk of diabetes in some breeds [35]. In addition, some polymorphisms have been identified in other immune response genes that contribute to the risk of diabetes in certain races [36].

Obesity increased significantly in dogs between 2011 and 2020 [1], being an important predisposing factor for insulin-resistant DM, as it interferes with glucose and insulin homeostasis. In this type of DM, there is a reversible state of insulin resistance due to impaired secretion, downregulation of insulin receptors, and post-receptor defects in the stimulation of systemic glucose transport, causing the degree of insulinemia to be directly correlated with the degree of insulin resistance [20]. In this sense, weight control is essential in the treatment of DM in dogs.

In diabetic dogs, control of hyperglycemia can be established with insulin therapy, diet, exercise, prevention or concomitant control of diseases with insulin antagonism, and the discontinuation of drugs that cause insulin resistance [2]. The treatment of an animal with diabetes requires the daily administration of insulin, making it necessary to create a routine and financial support, not only for the acquisition of insulin and consumption items, but also for the monitoring costs, exams, and hospitalization [31].

## 4. Pancreatitis

Pancreatitis is a disease characterized by inflammation in the exocrine pancreas; in dogs, it can be classified as acute or chronic. The triggering of pancreatitis occurs through intra-acinar cell activation of zymogens, leading to self-digestion of the gland, with cationic trypsinogen being the isoform responsible for cell death at the beginning of pancreatitis, which occurs due to the activation of NF-kB that may be involved in the progression of the local and systemic lesion [37]. It is noteworthy that the prolonged activation of NF-kB contributes to the infiltration of inflammatory cells, activation of stellate cells, loss of acinar cells, and fibrosis [38].

Histopathological examination is used as the gold standard for differentiating between the acute and chronic forms of the disease, in which, in acute pancreatitis, there is a mixed inflammatory infiltration of neutrophils, edema, and necrosis [39], which can be reversed, since the animals are treated. While in chronic pancreatitis there is progressive inflammation in the pancreas, in which there is loss of endocrine and exocrine cells, infiltration of inflammatory cells, fat replacement, activation of stellate cells, fibrosis, calcification and enlargement of the nerves [40].

It is worth mentioning that the histological changes are reversible in acute pancreatitis, however, when presented chronically, it can lead to compromise of endocrine and exocrine function until the development of DM and exocrine pancreatic insufficiency (EPI). Unlike the acute form, in which there is a high prevalence in the Miniature Schnauzer breed, the chronic form has a high prevalence in the English Cocker Spaniel breed [41].

As in humans, several factors can contribute to the development of pancreatitis, of which we can highlight: genetic predisposition, obesity, high-fat diet, alcoholism or smoking (humans only), medications, obstruction of the pancreatic ducts (stones, tumors, etc.), among others [41]. In dogs, in addition to the factors already discussed in humans, pancreatitis can be triggered by physical inactivity, idiopathic factors, toxins, breed, ischemia, reperfusion, hypercalcemia, infections (Toxoplasma), diet, trauma (crush or impact injuries to the abdomen), and endocrinopathies [42,43]. 

A study carried out by Bishop et al. [44] in miniature Schnauzer dogs, identified mutations in the *SPINK1* gene (serine protease inhibitor Kazal type 1, also known as pancreatic secreting trypsin inhibitor *PSTI*), as well as in humans, which may or may not be involved in the predisposition to the disease. Nevertheless, it has already been proven that the mutation alone does not cause spontaneous pancreatitis in humans, but it increases the susceptibility to alcoholic pancreatitis [45], so the mutation of this gene in dogs may also contribute to the development of pancreatitis when submitted to environmental changes.

The most common symptoms in dogs with pancreatitis are anorexia, vomiting, weakness, depression, abdominal pain (“prayer position”), and dehydration, however, symptoms may vary according to the severity of the pathology [46]. The mild stage of acute pancreatitis does not lead to organ failure, however, in more advanced stages it can cause multiple organ failure, systemic inflammatory response syndrome (SIRS), cardiovascular shock, multiple organ dysfunction syndrome or disseminated intravenous clotting [47,48,49,50].

The cause of pancreatitis remains unknown in most cases, so the treatment of pancreatitis remains almost exclusively supportive [51]. Treatment for both forms is similar and involves fluid therapy (IV), analgesia, nutritional treatment, antiemetics, gastro protectants, and antibiotics (prophylactic use for cases of severe acute pancreatitis), as well as treatments to deal with the loss of endocrine and exocrine function [43,47]. 

## 5. Extracellular Matrix Components

The extracellular matrix plays a fundamental role in the physical support for cells, in addition to enabling the regulation of several cellular processes, of which growth, migration, and differentiation, among others, can be highlighted [5,6,7]. It is worth mentioning that the ECM is composed of a network of macromolecules, which may vary according to each type of tissue.

The main constituents of the ECM are proteins such as collagens, elastin, fibronectin and laminin, glycoproteins, glycosaminoglycans (GAGs), and proteoglycans (PGs) [8,52]. The literature highlights that in the pancreatic ECM, specifically in the periphery of the islets, there is the presence of types I, II, III, IV, V and VI collagens, in addition to laminin, fibrin, and fibronectin, it is noteworthy that these last components involved in the process of cytoskeletal remodelling, contractility, and cell adhesion [53,54]. Collagens IV and VI are located mainly at the islet–exocrine interface beyond the islet basement membrane, playing the role of regulating fibronectin assembly by restricting cell–fibronectin interactions [55,56,57]. 

Fibronectins regulate various processes in the islets through interaction with integrin and non-integrin receptors on the islets and, and induce gene expression for differentiation makes for endocrine tissue, such as insulin 2, glucagon, *Pdx1*, and *Pax6* [54,58,59]. While laminin induces the expression of transcription factors and islet hormones such as Pdx1, insulin 1, insulin 2, glucagon, somatostatin, and GLUT−2, in addition to activating protein kinase B (Akt) and ERK (extracellular signal-regulated kinase), which act in the regulation of cellular metabolism [58,59]. Finally, fibrin can regulate the expression of α v β 3 integrin, preventing beta cell apoptosis [60,61].

A study carried out by Weber et al. [62] showed that the interaction between cells and ECM is necessary for beta cells to remain functional and avoid apoptosis, since ECM remodeling occurs during changes in physiological conditions, emphasizing the importance of studying matrix components that may be related to the development of diseases.

## 6. Extracellular Matrix Remodeling

ECM remodeling is a mechanism that results in the change of its composition (synthesis, degradation, and architecture) and the removal of some components (proteins, proteoglycans, and glycoproteins) for replacement by new components [63]. It is worth remembering that the processes of synthesis, degradation, reassembly, and modification of the ECM need to be controlled so that the tissue can maintain homeostasis and there is no deregulation of the components leading to the disease, in addition to aberrant remodeling that can trigger several pathological states such as cancer and fibrosis [64].

Thus, one of the main components that act in the degradation of the ECM are the metalloproteinases (MMPs), of which there are about 23 families in vertebrates, having a basic structure of three domains [65]. MMPs can degrade all ECM proteins, acting directly in organogenesis and morphogenesis. Under normal conditions, they have low activity, however, their activity is increased during repair or remodeling processes [63].

Most MMPs are secreted as zymogens and subsequently activated in the extracellular space, and their activation occurs through proteolytic cleavage or by modification of the thiol group by oxidation [63]. They can regulate the activity of other proteases, growth factors, cytokines, ligands, and cell surface receptors, in which their activity and that of other proteases ADAMS (A disintegrin and metalloproteinase proteases) and ADAMTs (A disintegrin and metalloproteinase with thrombospondin motifs) can be reversibly inhibited by tissue inhibitors of metalloproteinases (TIMPs) [66].

In addition to MMPs, there are other protease families such as ADAMs which are an integral membrane family and secreted glycoproteins with conserved protein domains [67]. They are divided into two subgroups: ADAMs and ADAMTs, which cleave ectodomains of transmembrane proteins, collagens, proteoglycans, deposit normal collagen fibrils in the ECM, and carry out cell signaling [64,66]. Another class of enzymes that can be highlighted are the meprins, which cleave ECM proteins such as collagen IV, nidogen, and fibronectin [68], in addition to acting indirectly in the regulation of ECM remodeling by activating other MMPs.

Proteolysis resulting from remodeling releases growth factors, inducing cell proliferation and migration, regulating organ morphogenesis [69], in addition to releasing biologically active molecules. This process requires precise regulatory mechanisms to avoid excessive ECM degradation and maintain tissue integrity.

When pancreatic cells undergo oxidative stress, there is the activation of MMPs and induction of remodeling fibrosis, resulting in loss of elasticity and stiffening of the ECM, affecting structures such as capillaries and exocrine ducts, in addition to impairing cell migration, contraction, and acinar loss and diffusion of molecules and hormones such as insulin. When we talk about pancreatic fibrosis, it is important to highlight the role of the fibrogenic pericyte cell and its relationship with the profibrogenic pancreatic stellate cell, which, when activated by incorrect pathways such as in oxidative stress, are differentiated into profibrotic cells capable of synthesizing type I and III collagens and fibronectin [70,71,72,73], accelerating the fibrosis process.

It has been shown that fibrosis via pancreatic ECM remodelling is a common pathway present in most chronic diseases, destroying tissue architecture, function, and organ failure as in type 2 DM in humans [74]. During the progression of diabetes, there is chronic infiltration of the pancreatic islets by mononuclear cells, and macrophages are considered the main agent causing the activation of cytotoxic T lymphocytes within the beta cells and the pancreatic islets [75].

When talking about DM and pancreatitis, two MMPs deserve special attention: MMP-2 and MMP-9 (gelatinases A and B, respectively), responsible for the degradation of gelatins, type III, IV, V, VII, X, and XI collagens, fibronectin, laminin, elastin, aggrecan, entactin, and vitronectin [63]. In the course of acute and chronic pancreatitis, the expression of MMP-9 is increased [76] as a result of the intra-acinar conversion of trypsinogen to trypsin, causing the activation of MMP−9.

The transforming growth factor−β (TGF−β) alters cell migration and regulates the expression of the ECM protein, in addition to inhibiting the expression and activity of MMP-2 leading to the expression of collagens and fibronectin [77]. However, during type 2 DM, there is an increase in TGF-β/Smad3 signaling, leading to β-cell apoptosis, causing glucose intolerance, β-cell dysfunction, decreased β-cell mass, and, consequently, insulin-resistant DM [78]. Therefore, it is extremely important to know the mechanisms of pancreatic ECM remodeling during pathological conditions.

## 7. Tissue Decellularization

Decellularization is a process that involves several steps to separate the extracellular matrix of the organ/tissue from the cellular components, leaving the ECM intact in terms of three-dimensional characteristics and biological properties [79]. The objective of this process is to efficiently remove all cellular and nuclear material, reducing any adverse effects on the composition, biological activity and mechanical integrity of the rest of the extracellular matrix [80]. Resulting in a scaffold that recapitulates the native characteristics of the organ or tissue serving as a basis for seeding with different cell types [79]. Among the studies found in the literature, the following models have already been used for pancreas decellularization (Figure 2): human [81,82,83,84,85,86,87], rabbit [88], murine [89,90,91], and swine [92,93,94,95,96,97]

Decellularization protocols typically involve a combination of chemical, physical, and biological treatments. The protocol normally starts with the lysis of the cell membrane through physical methods or ionic solutions, followed by the separation of the ECM components through biological treatment, aiming at the solubilization of the cytoplasmic and nuclear components using detergents, ending with the removal of cellular debris from the tissue [80].

Decellularization by chemical methods, using acids or bases, occurs through the use of chemical agents that act in the solubilization of the cytoplasmic components of the cells and the removal of nucleic acids such as DNA and RNA [80]. The most common acids are acetic and para-acetic, which demonstrate good cell removal capacity, however, they are aggressive to ECM, causing excessive loss of their properties [98]. The most used bases are calcium hydroxide, sodium sulfide, and sodium hydroxide [79]. However, despite their acceptance, these compounds eliminate growth factors such as glycosaminoglycans, resulting in loss of bioactivity [99]. Due to their characteristics, the aforementioned acids and bases are not universally used.

Thus, detergents represent the most used chemical agents in the decellularization process, acting in the solubilization of cell membranes [100], DNA separation from proteins, and being effective in removing cellular material from the treated tissue or organ [101].

The most used detergents (Figure 3) in this process are sodium dodecyl sulfate (SDS) and Triton X-100 [102]. SDS is an ionic and synthetic organic compound established in the field of tissue bioengineering, having several protocols mentioned in the literature for the decellularization of organs and tissues [102,103,104]. When compared to other detergents, SDS performs the complete removal of remaining nuclei and cytoplasmic proteins such as vimentin [105].

Among the ionic detergents, sodium deoxycholate (SDC) can also be highlighted, which is effective in removing remaining cells, but when compared to SDS, it tends to cause greater disruption in the architecture of the native tissue [80]. Triton X−100 is a non-ionic detergent composed of a hydrophilic polyethylene oxide chain and a lipophilic or hydrophobic aromatic hydrocarbon group. When compared to SDS, Triton X−100 can remove tissue cells and behaves less aggressively, thus making it interesting to use in thicker tissues [73]. Nonetheless, its effectiveness will depend on the tissue to be decellularized, the method used, and the decellularization protocol [80].

Decellularization by physical methods (Figure 4) involves several protocols that seek ways to remove the extracellular matrix through temperature protocols (freeze and thaw cycles), mechanical protocols (via shaking and immersion of samples), and pressure-based protocols [106]. Protocols that use temperature as a basis require several freeze–thaw cycles to become effective in cell removal [107]. Although this type of protocol is effective for cell removal, temperature changes can cause damage to the ECM structure [108], with this, the rate of temperature change must be controlled avoiding the formation of ice so as not to compromise the matrix [80]. Furthermore, the temperature protocol is recommended for the decellularization of simple structures, such as tendon- or cartilage-based organs, and is little applicable in organs with more complex structures, such as the pancreas, kidney, or liver [79].

Mechanical protocols involving sample agitation and immersion are often used in tissue bioengineering when samples are small and do not have vascular inlet and outlet, such as pre-cut cubes of parenchymal tissue, blood vessels [109], or bone fragments [110]. This method is normally used in conjunction with the chemical method to aid in cell lysis and removal and can be applied through a magnetic stir plate, orbital shaker, or low-profile roller [80]. This protocol makes it possible to change the duration of the protocol or the agitation force, allowing the adjustment according to the tissue density. However, there is a need for a decellularization process through a homogeneous process, due to some organs presenting higher degradation of the extracellular matrix at the external region than the inner part of the sample [79]. Even so, physical decellularization protocols need to be combined with chemical protocols as they are usually insufficient for complete decellularization to occur [80].

In decellularization by biological treatment, cell membranes and the bonds responsible for intercellular and extracellular connections are disrupted [80]. The protocols involve the use of enzymatic and non-enzymatic biological agents that are capable of removing unwanted cell residues or constituents from the extracellular matrix [79]. DNases and RNases are the most used enzymes, as they can cleave nucleic acid sequences, helping to eliminate nucleotides after cell lysis [111]. Non-enzymatic agents are represented by chelators for divalent ions such as Ethylenediaminetetraacetic acid (EDTA) and Ethylene glycol-bis(2-aminoethylether)-N,N,N′,N′-tetraacetic acid (EGTA) which, through sequestering metal ions, separate cells from the extracellular matrix [79]. However, these agents are not effective when used alone, so they are often added to multi-step protocols [112,113,114].

## 8. Tissue Recellularization

The recellularization process consists of repopulating the scaffolds with cultured cells. Recellularization requires three different cell groups, belonging to parenchymal (effective organ function), vascular (providing adequate blood flow), and supportive (sustaining parenchymal and vascular cells) types [79]. For the success of this process, the scaffold must have its composition and three-dimensional structure preserved, adequate cell types, an efficient cell seeding method, and a cell culture environment similar to the physiological one [115].

There are reports in the literature on the recellularization of the pancreas with different cell types, among them: mesenchymal stem cells from human placenta (hPL-MSC) [88], human vascular endothelial cells (HUVEC) [116,117,118,119], pancreatic islet cells of murine (mouse and rat), human and swine [91,93,120,121,122,123,124], bone marrow-derived mesenchymal stromal cells (BMSCs) [125], fibroblasts [126], insulinoma cells (INS-1E) [89,94], endothelial progenitor cells (EPCs) [127], isolated human fetal pancreas cells (hFPSC) [96], and induced pluripotent stem cells (iPSCs) [128].

Fetal progenitor cells have a high capacity to proliferate and become adult cells from the tissue they originate, being successfully tested in the recellularization process [129,130,131]. Rat fetal progenitor cells were able to perform physiological functions such as gas exchange in recellularized lungs [132]. Even though these cells show evidence of the ability to restore cell function in decellularized scaffolds, this cell type is not desirable for clinical application, due to ethical concerns related to obtaining these cells, since they are derived from fetuses [133].

Adult cells are candidates to be used in the recellularization process, as they have the advantages of already being differentiated and expressing the genes necessary to perform the functions of the tissue of origin [134,135,136]. These cells can be obtained by biopsy from the patient’s organ or donor organs [137]. Due to the low proliferative capacity of these cells, their application is limited [133].

Another group of cells that can be used in recellularization are stem cells, which have a high proliferative capacity and can differentiate into desired cell types, in addition to being the cell types most used in recellularization assays due to their advantages over other cell types [138].

There are several reports of different procedures and cells used in seeding, perfusion, and injections [139], however, clinical tests after the recellularization process are still poorly published and there is a need for more studies related to the host response after implantation [140].

## 9. Use of Pancreatic ECM as a Therapeutic Possibility for the Treatment of DM and Pancreatitis

It is known that so far there is no possibility of a cure for diabetes mellitus and pancreatitis. In this sense, several studies have been developed in search of a therapeutic approach for reducing the use of medicines. Among them, the pancreatic islet transplantation can be highlighted, which gives the possibility of glycemic control, however, there is a need for the use of immunosuppressants by patients in addition to the gradual loss of function of the islets. [141].

It has already been demonstrated that ECM plays an important role in addition to influencing cellular function. The use of decellularized organs has emerged in recent years in tissue engineering and is a promising technique, since these tissues have a great advantage due to the reduction of immunogenicity when all cellular contents are removed, serving as a support for transplanted cells. In this context, the use of the pancreatic extracellular matrix would be ideal for islet transplantation, as well as other cell types, since its ECM presents the necessary components for the regulation of cellular processes, allowing cellular survival. [142]. 

It is worth mentioning that some studies using islets seeded in pancreatic scaffolds have already demonstrated that in vitro cellular function and survival were maintained [87,91,93,143,144]. Nevertheless, there is a need for further in vivo studies. Studies being carried out by our research group using canine and rat mesenchymal stem cells seeded in canine pancreas scaffolds, allowed us to observe the survival and proliferation of both cells, which could be used in the future as an option therapeutics in veterinary medicine for the treatment of pancreatic diseases [145,146].

## 10. Conclusions

Dogs play an important and historical role in the research of diabetes mellitus, considering that it was in the canine species that this disease was produced for the first time experimentally, in addition to the species having many inherited diseases that arise naturally and mimic those observed in humans. It makes dogs an experimental model suitable for the study of the pancreatic extracellular matrix aiming at a better understanding of both diabetes mellitus and pancreatitis, since there are no reports in the literature regarding the composition of the ECM of these animals, as well as studies related to matrix components that are involved in triggering these diseases.

## Figures and Tables

**Figure 1 animals-13-00684-f001:**
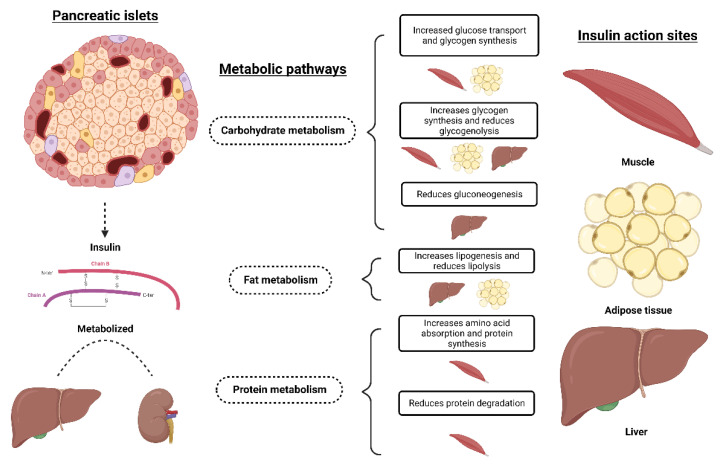
Influence of insulin action on different metabolic pathways and sites of action. Created with BioRender.com.

**Figure 2 animals-13-00684-f002:**
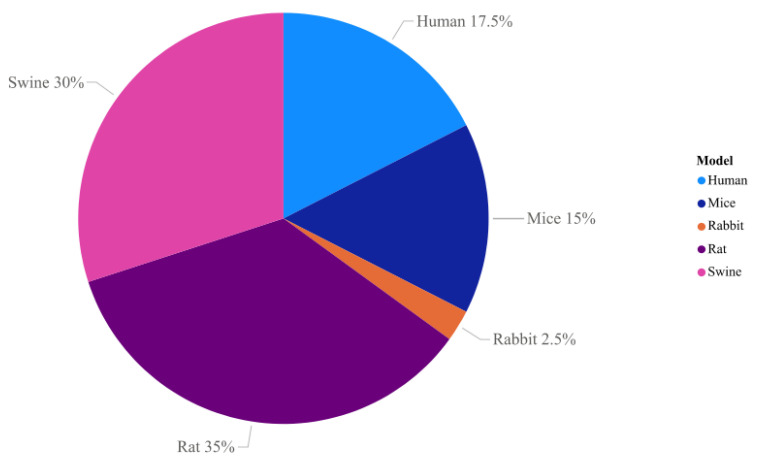
Models found in the PubMed database for pancreas decellularization between 2009 and 2021. The rat is the most used animal so far, representing 35% of the studies, followed by the swine model (30%), humans present about 17.5% of the data described in the literature, soon after it can be highlighted the mice (15%), while the least used species so far are rabbits (2.5%).

**Figure 3 animals-13-00684-f003:**
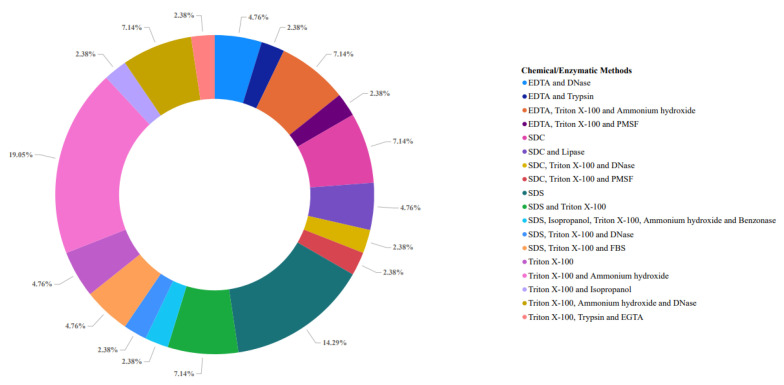
Chemical/enzymatic methods for pancreas decellularization found in the PubMed database between 2009 and 2021. The most used method to date includes the use of Triton X−100 and ammonium hydroxide (17.39%), followed by the use of only SDS (13.04%), while the rest of the studies addressed the use of different types of detergents (ionic and non-ionic) simultaneously in the decellularization process.

**Figure 4 animals-13-00684-f004:**
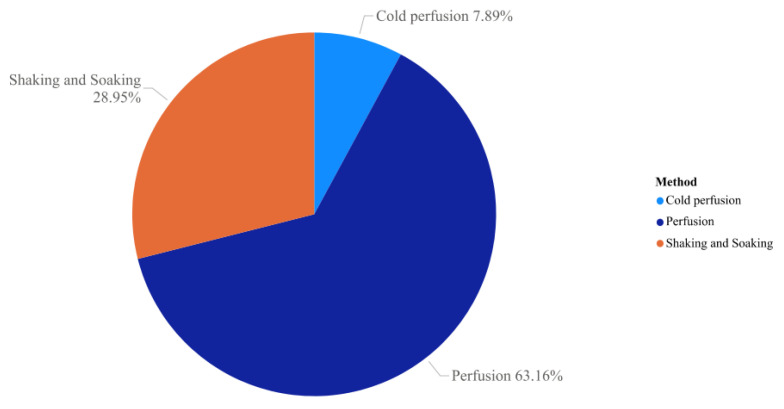
Physical methods used in pancreas decellularization found in the PubMed database between 2009 and 2021. The most used physical method is perfusion (63.16%), followed by agitation and immersion (28.95%) which is classified as a mechanical protocol and cold perfusion (7.89%) that uses temperature to assist in the process.

## Data Availability

Not applicable.

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
