# Peer review of "The Canine Pancreatic Extracellular Matrix in Diabetes Mellitus and Pancreatitis: Its Essential Role and Therapeutic Perspective"

_animals, 2023, doi:10.3390/ani13040684_

Round 1

Reviewer 1 Report

Dear authors,

I consider this initiative of yours to be very valid and with enormous potential. However, I believe that there will have to be an additional work of contextualizing the ideas in order to give substance to the title of the work.

I believe that it is also not very clear to the reader of your work how the use of pancreatic extracellular matrix, as a scaffold, can be useful in the treatment of these two diseases.

In my opinion, a point should be created with the goals to be achieved in the treatment of these two diseases using a cell-seeded pancreatic extracellular matrix scafold.

In short, the work lacks a better characterization of the extracellular matrix of the healthy pancreas as well as the alterations that appear in this matrix in the two diseases under analysis. It is also important to better clarify the role of the extracellular matrix in the new treatments of regenerative medicine in the context of these two diseases, at which point the work is practically omitted.

3. Diabetes Mellitus

Line 199 -Please write the breed names starting with capital letters as you have done elsewhere in the text.

4. Pancreatitis

Line 256 - Contrary to what happens in man, pancreatitis in animals is predominantly aseptic. The indication of antibiotic therapy should be reviewed and contextualized.

Lines 257 to 259 - "It is worth mentioning that the pancreas has a high capacity for functional reserve, causing pancreatitis to develop only if more than 80 – 90% of the functional mass of the pancreas is compromised." I am afraid that this sentence is related with Exocrine pancreatic insufficiency and not with pancreatitis. Reference is due.

5. Extracellular Matrix Components

Taking into account the objective of the work, a greater detail in the description of the extracellular matrix of the healthy pancreas is justified. Furthermore, at this point, we could not find any reference to the alterations existing in the extracellular matrix of the pancreas in the diabetic animal and in the animal with pancreatitis.

6. Extracellular Matrix Remodeling

Line 329 - squamous cell carcinomas

Author Response

Thanks for the considerations, a better description of the compliant pancreatic ECM, we also highlight the healthy literature characterizing an extracellular matrix deepened by diseases, and there is a need for studies regarding this. In this sense, we added a topic entitled "Use of pancreatic ECM as a therapeutic possibility for the treatment of DM and Pancreatitis" emphasizing the importance of using pancreatic ECM and its possible applications in regenerative medicine, in addition to the progress of studies that seek its use mainly for the treatment of Diabetes Mellitus.

Topic added in text (Line 482 to 504)

“9. Use of pancreatic ECM as a therapeutic possibility for the treatment of DM and Pancreatitis

It is known that so far there is no possibility of a cure for Diabetes Mellitus and Pancreatitis, in this sense, several studies have been developed in search of a therapeutic possibility for reducing the use of medicines. Among the pancreatic islet transplantation can be highlighted, which gives the possibility of glycemic control, however, there is a need for the use of immunosuppressants by patients in addition to the gradual loss of function of the islets. [138].

It has already been demonstrated that ECM plays an important role in addition to influencing cellular function. The use of decellularized organs has emerged in recent years in Tissue Engineering and is a promising field since these tissues have a great advantage due to the reduction of immunogenicity when all cellular contents are removed, serving as a support for transplanted cells. In this context, the use of the pancreatic extracellular matrix would be ideal for use in islet transplantation, as well as other cell types, since its ECM presents the necessary components for the regulation of cellular processes, allowing cellular survival. [139].

It is worth mentioning that some studies using islets seeded in pancreatic scaffolds have already demonstrated that in vitro cellular function and survival were maintained [84,88,90,140,141], however, there is a need for further in vivo studies. Studies being carried out by our research group using mesenchymal stem cells from dogs and rats seeded in biological scaffolds derived from dog pancreas allowed the survival and cellular proliferation of both cells, which could be used in the future as an option therapeutics in veterinary medicine for the treatment of pancreatic diseases [142,143].”

  1. Diabetes Mellitus

Line 199 -Please write the breed names starting with capital letters as you have done elsewhere in the text.

Thanks for the consideration, the names of the races have been changed to capital letters.

Before:

“The poodle, pinscher and miniature schnauzer, dachshund and beagle breeds”

After:

“The Poodle, Pinscher and Miniature Schnauzer, Dachshund and Beagle breeds”

  1. Pancreatitis

Line 256 - Contrary to what happens in man, pancreatitis in animals is predominantly aseptic. The indication of antibiotic therapy should be reviewed and contextualized.

Thanks for the considerations, although the development of pancreatitis in dogs has a less in-depth elucidation than in humans, which is usually developed due to obesity, smoking, or alcohol for example. There are reports in the literature that this disease in dogs can be triggered by a series of facts from a sedentary lifestyle, obesity, idiopathic factors, medications, toxins, race, gender, ischemia, reperfusion, hypercalcemia, duct obstruction, infections, diet, in addition to trauma such as crushing or impact injuries to the abdomen (Santos and Alessi, 2016).

Regarding antibiotic therapy, its use is recommended in Veterinary Medicine as a prophylactic measure in cases of severe acute pancreatitis (Nelson and Couto, 2015). Accordingly, the following changes were made:

Before (Line 240 to 245):

“As in humans, several factors can contribute to the development of pancreatitis, of which we can highlight: genetic predisposition, obesity, high-fat diet, alcoholism or smoking (humans), medications, obstruction of the pancreatic ducts, among others [41]. A study carried out by Bishop et al. [42] in miniature Schnauzer dogs, identified mutations in the SPINK1 gene (pancreatic secreting trypsin inhibitor), as well as in humans, which may or may not be involved in the predisposition to the disease.”

After (Line 241 to 250):

“As in humans, several factors can contribute to the development of pancreatitis, of which we can highlight: genetic predisposition, obesity, high-fat diet, alcoholism or smoking (humans), medications, obstruction of the pancreatic ducts, among others [41]. In dogs, pancreatitis can be triggered by physical inactivity, obesity, idiopathic factors, some medications, toxins, breed, ischemia, reperfusion, hypercalcemia, obstruction of the pancreatic ducts (stones, tumors, etc.), infections (Toxoplasma), diet, trauma (crush or impact injuries to the abdomen, endocrinopathies [42,43]. A study carried out by Bishop et al. [44] in miniature Schnauzer dogs, identified mutations in the SPINK1 gene (pancreatic secreting trypsin inhibitor), as well as in humans, which may or may not be involved in the predisposition to the disease.”

Before (Line 255 to 257):

“Treatment for both forms is similar and involves fluid therapy (IV), analgesia, nutritional treatment, antiemetics, gastro protectants, antibiotics, as well as treatments to deal with the loss of endocrine and exocrine function [45].”

After (Line 260 to 263):

“Treatment for both forms is similar and involves fluid therapy (IV), analgesia, nutritional treatment, antiemetics, gastro protectants and antibiotics (prophylactic use for cases of severe acute pancreatitis), as well as treatments to deal with the loss of endocrine and exocrine function [43,47].”

Lines 257 to 259 - "It is worth mentioning that the pancreas has a high capacity for functional reserve, causing pancreatitis to develop only if more than 80 – 90% of the functional mass of the pancreas is compromised." I am afraid that this sentence is related with Exocrine pancreatic insufficiency and not to pancreatitis. Reference is due.

Thanks for the consideration, a literature review was performed and the excerpt was removed from the review since the information corresponds to exocrine pancreatic insufficiency.

  1. Extracellular Matrix Components

Taking into account the objective of the work, greater detail in the description of the extracellular matrix of the healthy pancreas is justified. Furthermore, at this point, we could not find any reference to the alterations existing in the extracellular matrix of the pancreas in the diabetic animal and in the animal with pancreatitis.

Thanks for your comments, the following changes were made to the text:

Before (Line 268 to 270):

“The literature highlights that in the pancreatic ECM, specifically in the periphery of the islets, there is the presence of collagen types I, III, IV, V and VI, in addition to laminin and fibronectin [47].”

After (Line 272 to 285):

“The literature highlights that in the pancreatic ECM, specifically in the periphery of the islets, there is the presence of collagen types I, II, III, IV, V and VI, in addition to laminin, fibrin and fibronectin, it is noteworthy that these last components involved in the process of cytoskeletal remodeling, contractility and cell adhesion [49,50]. Collagens IV and VI are located mainly at the islet-exocrine interface beyond the islet basement membrane, playing the role of regulating fibronectin assembly by restricting cell-fibronectin interactions [51–53].

Fibronectins regulate various processes in the islets through interaction with integrin and non-integrin receptors on the islets. endocrine tissue, such as insulin 2, glucagon, Pdx1, and Pax6 [50,54,55], while laminin induces the expression of transcription factors and islet hormones such as: Pdx1, insulin 1, insulin 2, glucagon, somatostatin and GLUT-2, in addition to activating protein kinase B (Akt) and ERK (Extracellular signal-regulated kinase), which act in the regulation of cellular metabolism [54,55]. Finally, fibrin can regulate the expression of α v β 3 integrin, preventing beta cell apoptosis [56,57].”

Regarding the description of ECM affected by Diabetes and Pancreatitis, it is worth mentioning that there are no studies in the literature describing this type of alteration. We emphasize that at the moment we are developing an in-depth study of the alterations of these extracellular matrix components in the canine pancreas affected by Diabetes Mellitus and Pancreatitis.

  1. Extracellular Matrix Remodeling

Line 329 - squamous cell carcinomas

Thank you for your comments, we have made the following change to the text:

Before (Line 329):

“squamous cancer”

After (Line 344):

“squamous cell carcinomas”

Reviewer 2 Report

The review entitled: “The canine pancreatic extracellular matrix in Diabetes Mellitus And Pancreatitis: it’s essential role and therapeutical perspective on the pathogenesis“ by Bruna Tássia dos Santos Pantoja, Rafael Cardoso Carvalho, Maria Angelica Miglino and Ana Claudia Oliveira Carreira, highlight the important role to have used of dog model to study several pathologies, like diabetes mellitus, since this model present phenotypically similar development in humans. Authors focus the attention on the importance to study extracellular matrix (ECM) remodeling in canine’s pancreas, triggered by pancreatitis to diabetes evolution. Authors has done very extensive and exhaustive literature search of mechanism involved in remodeling of ECM in several disease in different animal models and in humans. Authors underline the fact the in literature is not reported until now the characterization of canine’s pancreas ECM even though development of diseases lead to ECM remodeling process. Despite the interesting overview of pancreatitis-diabetes and ECM remodeling, Authors are invited to adjust minor things as follow:

-        Line 8: alline “Paulo” with the line above and under
-        Line 328: correct “causiem ng” vs. “causing”
-        Line 374: correct “in” vs. “In”

Authors are strongly invited to reconsider to rewrite in proper form the paragraphs “Simple Summary” and “Abstract” rearranging and eliminate the repeated fraises. Furthermore, the paragraph “Conclusion” is confuse and it must be clarify the meaning using, also, the proper punctuation.

Author Response

-        Line 8: alline “Paulo” with the line above and under

The text has been aligned as per the recommendations

Before:

1  Department of Surgery, School of Veterinary Medicine and Animal Science, University of São Paulo, São

Paulo, Brazil, 05508-270; [email protected] (B.T.S.P.); [email protected] (M.A.M.)

After:

1  Department of Surgery, School of Veterinary Medicine and Animal Science, University of Sao Paulo, Sao Paulo, Sao Paulo, Brazil, 05508-270; [email protected] (B.T.S.P.); [email protected] (M.A.M.); [email protected] (A.C.O.C.)

-        Line 328: correct “causiem ng” vs. “causing”

The correction was made as requested.

Before:

“causiem ng” the activation of MMP-9

After:

“causing” the activation of MMP-9

-        Line 374: correct “in” vs. “In”

A correction was made as requested.

Before:

“in” the literature for

After:

“In” the literature for

Authors are strongly invited to reconsider to rewrite in proper form the paragraphs “Simple Summary” and “Abstract” rearranging and eliminate the repeated fraises. Furthermore, the paragraph “Conclusion” is confuse and it must be clarify the meaning using, also, the proper punctuation.

Thanks for the consideration, changes have been made to the simple summary, as requested:

Before:

Simple Summary: The dog is considered an animal model for the study of several diseases that occur in humans, since they present phenotypically similar development. Among the diseases that dogs are used as a study model, Diabetes Mellitus and Pancreatitis can be highlighted, which are diseases that affect the endocrine and exocrine portion of the pancreas, with a high prevalence in companion animals in addition to high social cost and great impact. in mortality and morbidity. The objective of this work was to highlight the importance of using the extracellular matrix of the canine pancreas as a model for the study of Diabetes Mellitus and Pancreatitis. There are no reports in the literature characterizing the extracellular matrix of the canine pancreas, only that of mice, swine, humans, and rabbits. The extracellular matrix performs several functions (physical support and regulation of cellular processes), being composed mainly of proteins, glycoproteins, glycosaminoglycans and proteoglycans. It has already been demonstrated that pathological conditions such as the triggering of endocrine and exocrine diseases cause the extracellular matrix to be modified through the ECM remodeling process, highlighting the importance of in-depth study of the changes resulting from these diseases.

After:

Simple Summary: The dog is considered an animal model for the study of several diseases that occur in humans, since they present phenotypically similar development. Among the diseases that dogs are used as a study model, Diabetes Mellitus and Pancreatitis can be highlighted, which are diseases that affect the endocrine and exocrine portion of the pancreas, with a high prevalence in companion animals in addition to high social cost and great impact in mortality and morbidity. The aim of this work was to highlight the importance of using the dog as a model for the study of changes in the pancreatic extracellular matrix when affected by Diabetes Mellitus and Pancreatitis. The extracellular matrix performs several functions (physical support and regulation of cellular processes), being composed mainly of proteins, glycoproteins, glycosaminoglycans and proteoglycans. It is worth mentioning that there are no studies characterizing the canine pancreatic extracellular matrix, as well as studies related to the matrix components involved in the triggering of these diseases. It is known that pathological conditions such as the triggering of endocrine and exocrine diseases cause the extracellular matrix to be modified through the ECM remodeling process, emphasizing the importance of studying the changes resulting from these diseases.

Round 2

Reviewer 1 Report

Dear authors

Simple Summary

Line 18 - and exocrine portion of the pancreas,respectively,..

Line 19 - in addition to high social cost ??? - What do you understand by high social cost in this context?

Line 43  - Keywords: Decellularization; ECM remodeling; Endocrine and Exocrine Diseases;  Pancreas; Scaffolds; Tissue Bioengineering.

3. Diabetes Mellitus

Line 191 - sarcopenia

4. Pancreatitis

Line 225 to 226 - in the progression. of the local and systemic lesion [37].

Lines 241 to 247 - Please better harmonize text

As in humans, several factors can contribute to the development of pancreatitis, of which we can highlight: genetic predisposition, obesity, high-fat diet, alcoholism or smoking (humans), medications, obstruction of the pancreatic ducts, among others [41]. In dogs, pancreatitis can be triggered by physical inactivity, obesity, idiopathic factors, some medications, toxins, breed, ischemia, reperfusion, hypercalcemia, obstruction of the pancreatic ducts (stones, tumors, etc.), infections (Toxoplasma), diet, trauma (crush or impact injuries to the abdomen, endocrinopathies [42,43].

As well for next phrases related with genetic issue. Please write only once and in an integrated way on the subject "genetics and pancreatitis". At the moment we find several paragraphs dealing with the same subject.

study carried out by Bishop et al. [44]  in miniature Schnauzer dogs, identified mutations in the SPINK1 gene (pancreatic secreting trypsin inhibitor), as well as in humans, which may or may not be involved in the  predisposition to the disease.

However, it has already been proven that the mutation alone does not cause spontaneous pancreatitis in humans, however, it increases the susceptibility to alcoholic pancreatitis [45], so the mutation of this gene in dogs may also contribute to the development of pancreatitis when submitted to environmental changes.

Lines 255 to 259 - The most common symptoms in dogs with pancreatitis are anorexia, depression, abdominal pain, and dehydration, however, symptoms may vary according to the severity of the pathology. The mild stage of acute pancreatitis does not lead to organ failure, however, in more advanced stages it can cause multiple organ failure among other complications [46], such as cardiovascular shock and disseminated intravenous coagulation [47].

Very outdated bibliographic references, please reconsider updating this information, notably referring to the clinical sign of vomiting and complications such as systemic inflammatory response syndrome, DIC and multiorgan dysfunction syndrome.

Line 260 to 263 - Pancreatitis is mostly asseptic in dogs.

I suggest these references:

Journal of Small Animal Practice (2015) 56, 27–39 DOI: 10.1111/jsap.12296. C. Mansfield and T. Beths. Management of acute pancreatitis in dogs: a critical appraisal with focus on feeding and analgesia

Canine and feline gastrenterology. Chapter 60. Edited by Robert J. Washabau and Michael J. Day. https://doi.org/10.1016/C2009-0-34969-7

6. Extracellular Matrix Remodeling

Line 299 to 301 - Thus, one of the main components that act in the degradation of the ECM are the metalloproteinases (MMPs), of which there are about 23 families in vertebrates, having a basic structure of three domains [61].

Line 310 to 311 - ADAMs (A disintegrin and metalloproteinase proteases)

Line 313 - ADAMTs (A disintegrin and metalloproteinase with thrombospondin motifs),

Line 330 - capable of synthesizing type I and collagen types. III and fibronectin

Line 343 to 344 - Trypsin is known to have a particular role in squamous cell carcinomas, in which MMPs are activated by trypsin-2, facilitating invasion and metastasis [73]. What do you intend to demonstrate by referring to this tumor?

Line 369 -Decellularization protocols typically involve a combination of  physical, chemical 

You must reorganize the entire text between lines 369 and 444, linking the subjects in a fluid and logical way and harmonizing the discourse of various authors in a new text that is your text.

As in the first sentence, you begin your speech with the physical methods, you should exhaust the subject of physical methods and only then go on to enumerate other methods.

Line 369 -Decellularization protocols typically involve a combination of  physical, chemical

Ex: These two sentences address the same subject. You must integrate the two sentences into one.

Line 371 to 373 - followed by the separation of the ECM components through biological treatment, aiming at the solubilization of the cytoplasmic and nuclear components using detergents,

Line 375 to 377 - Decellularization by chemical methods, using acids or bases, occurs through the use of chemical agents that act in the solubilization of the cytoplasmic components of the cells and in the removal of nucleic acids such as DNA and RNA [77].

Line 389 to 390 - In the literature for decellularization of organs and tissues [99–101]. ??????????????

8. Tissue Recellularization

These cells can be obtained by biopsy from the patient's organ or donor organs [134]. However, the biopsy can generate a genetic defect, making them unviable and, ...

Could you please explain how biopsy cause a genetic defect?

Author Response

Dear authors

Simple Summary

Line 18 - and exocrine portion of the pancreas,respectively,..

Thanks for the consideration, modification was made.

Before:

“diseases that affect the endocrine and exocrine portion of the pancreas, with a high prevalence in”

After:

“diseases that affect the endocrine and exocrine portion of the pancreas, respectively, with a high “

Line 19 - in addition to high social cost ??? - What do you understand by high social cost in this context?

High cost in relation to the use of medicines, consultations, diets, etc.

Line 43  - Keywords: Decellularization; ECM remodelling; Endocrine and Exocrine Diseases;  Pancreas; Scaffolds; Tissue Bioengineering.

Thanks for the considerations, the modifications have been made.

Before:

“Keywords: ECM remodeling; Endocrine and Exocrine Diseases; Decellularization; Scaffolds; Tissue Bioengineering.”

After:

“Keywords: Decellularization; ECM remodeling; Endocrine and Exocrine Diseases; Pancreas; Scaffolds; Tissue Bioengineering.”

  1. Diabetes Mellitus

Line 191 - sarcopenia

Thank you for your consideration, the text has been modified.

Before:

“thus causing weight loss, hepatomegaly and muscle wasting”

After:

“thus causing weight loss, hepatomegaly and sarcopenia”

  1. Pancreatitis

Line 225 to 226 - in the progression. of the local and systemic lesion [37].

Thank you for your consideration, the text has been modified.

Before:

“the progression. of the local and systemic lesion [37].”

After:

“in the progression of the local and systemic lesion [37].”

Lines 241 to 247 - Please better harmonize text

As in humans, several factors can contribute to the development of pancreatitis, of which we can highlight: genetic predisposition, obesity, high-fat diet, alcoholism or smoking (humans), medications, obstruction of the pancreatic ducts, among others [41]. In dogs, pancreatitis can be triggered by physical inactivity, obesity, idiopathic factors, some medications, toxins, breed, ischemia, reperfusion, hypercalcemia, obstruction of the pancreatic ducts (stones, tumors, etc.), infections (Toxoplasma), diet, trauma (crush or impact injuries to the abdomen, endocrinopathies [42,43].

As well for next phrases related with genetic issue. Please write only once and in an integrated way on the subject "genetics and pancreatitis". At the moment we find several paragraphs dealing with the same subject.

study carried out by Bishop et al. [44]  in miniature Schnauzer dogs, identified mutations in the SPINK1 gene (pancreatic secreting trypsin inhibitor), as well as in humans, which may or may not be involved in the  predisposition to the disease.

However, it has already been proven that the mutation alone does not cause spontaneous pancreatitis in humans, however, it increases the susceptibility to alcoholic pancreatitis [45], so the mutation of this gene in dogs may also contribute to the development of pancreatitis when submitted to environmental changes.

Thank you for your consideration, the following changes have been made to the text:

Before:

“As in humans, several factors can contribute to the development of pancreatitis, of which we can highlight: genetic predisposition, obesity, high-fat diet, alcoholism or smoking (humans), medications, obstruction of the pancreatic ducts, among others [41]. In dogs, pancreatitis can be triggered by physical inactivity, obesity, idiopathic factors, some medications, toxins, breed, ischemia, reperfusion, hypercalcemia, obstruction of the pancreatic ducts (stones, tumors, etc.), infections (Toxoplasma), diet, trauma (crush or impact injuries to the abdomen, endocrinopathies [42,43]. A study carried out by Bishop et al. [44] in miniature Schnauzer dogs, identified mutations in the SPINK1 gene (pancreatic secreting trypsin inhibitor), as well as in humans, which may or may not be involved in the predisposition to the disease.”

After:

“As in humans, several factors can contribute to the development of pancreatitis, of which we can highlight: genetic predisposition, obesity, high-fat diet, alcoholism or smoking (humans), medications, obstruction of the pancreatic ducts (stones, tumors, etc.), among others [41]. In dogs, in addition to the factors already discussed in humans, pancreatitis can be triggered by physical inactivity, idiopathic factors, toxins, breed, ischemia, reperfusion, hypercalcemia, infections (Toxoplasma), diet, trauma (crush or impact injuries to the abdomen) and endocrinopathies [42,43].

A study carried out by Bishop et al. [44] in miniature Schnauzer dogs, identified mutations in the SPINK1 gene (pancreatic secreting trypsin inhibitor), as well as in humans, which may or may not be involved in the predisposition to the disease. However, it has already been proven that the mutation alone does not cause spontaneous pancreatitis in humans, however, it increases the susceptibility to alcoholic pancreatitis [45], so the mutation of this gene in dogs may also contribute to the development of pancreatitis when submitted to environmental changes.”

Lines 255 to 259 - The most common symptoms in dogs with pancreatitis are anorexia, depression, abdominal pain, and dehydration, however, symptoms may vary according to the severity of the pathology. The mild stage of acute pancreatitis does not lead to organ failure, however, in more advanced stages it can cause multiple organ failure among other complications [46], such as cardiovascular shock and disseminated intravenous coagulation [47].

Very outdated bibliographic references, please reconsider updating this information, notably referring to the clinical sign of vomiting and complications such as systemic inflammatory response syndrome, DIC and multiorgan dysfunction syndrome.

Thanks for the consideration. The authors recognize that the reference is old, especially reference number 46. We performed a new search for updated references and authors who study the topic continue to cite the oldest references, with no need to change the references in our text.

We took into account the fact that the clinical signs of vomiting, systemic inflammatory response syndrome, DIC, and multiorgan dysfunction syndrome were added to the text. With that, our text looked like this after the changes:

Before:

“The most common symptoms in dogs with pancreatitis are anorexia, depression, abdominal pain, and dehydration, however, symptoms may vary according to the severity of the pathology. The mild stage of acute pancreatitis does not lead to organ failure, however, in more advanced stages it can cause multiple organ failure among other complications [46], such as cardiovascular shock and disseminated intravenous coagulation [47].”

After:

“The most common symptoms in dogs with pancreatitis are anorexia, vomiting, weakness, depression, abdominal pain (“prayer position”), and dehydration, however, symptoms may vary according to the severity of the pathology [46]. The mild stage of acute pancreatitis does not lead to organ failure, however, in more advanced stages it can cause multiple organ failure, systemic inflammatory response syndrome (SIRS), cardiovascular shock, multiple organ dysfunction syndrome or disseminated intravenous clotting [47–50].”

Line 260 to 263 - Pancreatitis is mostly asseptic in dogs.

I suggest these references:

Journal of Small Animal Practice (2015) 56, 27–39 DOI: 10.1111/jsap.12296. C. Mansfield and T. Beths. Management of acute pancreatitis in dogs: a critical appraisal with focus on feeding and analgesia

Canine and feline gastrenterology. Chapter 60. Edited by Robert J. Washabau and Michael J. Day. https://doi.org/10.1016/C2009-0-34969-7

Thank you for your comments, we have made the following changes:

Before:

“Treatment for both forms is similar and involves fluid therapy (IV), analgesia, nutritional treatment, antiemetics, gastro protectants and antibiotics (prophylactic use for cases of severe acute pancreatitis), as well as treatments to deal with the loss of endocrine and exocrine function [43,47].”

After:

“The cause of pancreatitis remains unknown in most cases, so the treatment of pancreatitis remains almost exclusively supportive [51]. Treatment for both forms is similar and involves fluid therapy (IV), analgesia, nutritional treatment, antiemetics, gastro protectants and antibiotics (prophylactic use for cases of severe acute pancreatitis), as well as treatments to deal with the loss of endocrine and exocrine function [43,48].”

  1. Extracellular Matrix Remodeling

Line 299 to 301 - Thus, one of the main components that act in the degradation of the ECM are the metalloproteinases (MMPs), of which there are about 23 families in vertebrates, having a basic structure of three domains [61].

The following change was made:

Before:

“Thus, one of the main components that act in the degradation of the ECM is the metalloproteinases (MMPs), of which there are about 23 families in vertebrates, having a basic structure of three domains [61].”

After:

“Thus, one of the main components that act in the degradation of the ECM are the metalloproteinases (MMPs), of which there are about 23 families in vertebrates, having a basic structure of three domains [65].”

Line 310 to 311 - ADAMs (A disintegrin and metalloproteinase proteases)

The following change was made:

Before:

“In addition to MMPs, there are other protease families such as ADAMs (disintegrin and metalloprotease proteases) which are an integral membrane family and secreted glycoproteins with conserved protein domains [63].”

After:

“In addition to MMPs, there are other protease families such as ADAMs (A disintegrin and metalloproteinase proteases) which are an integral membrane family and secreted glycoproteins with conserved protein domains [67].”

Line 313 - ADAMTs (A disintegrin and metalloproteinase with thrombospondin motifs),

The following change was made:

Before:

“They are divided into two subgroups: ADAMs and ADAMTs (disintegrin and metalloproteinase with thrombospondin motifs),”

After:

“They are divided into two subgroups: ADAMs and ADAMTs (A disintegrin and metalloproteinase with thrombospondin motifs),”

Line 330 - capable of synthesizing type I and collagen types. III and fibronectin

The following change was made:

Before:

“capable of synthesizing type I and collagen types. III and fibronectin [66–69],”

After:

“capable of synthesizing type I and collagen types III and fibronectin [70–73]”

Line 343 to 344 - Trypsin is known to have a particular role in squamous cell carcinomas, in which MMPs are activated by trypsin-2, facilitating invasion and metastasis [73]. What do you intend to demonstrate by referring to this tumor?

Thanks for the comments. We revised the text and the excerpt was removed from the review.

Before:

“When talking about DM and pancreatitis, two MMPs deserve special attention - MMP-2 and MMP-9 (gelatinases A and B, respectively), responsible for the degradation of gelatins, collagens III, IV, V, VII, X and XI, fibronectin, laminin, elastin, aggrecan, entactin and vitronectin [59]. In the course of acute and chronic pancreatitis, the expression of MMP-9 is increased [72] as a result of the intra-acinar conversion of trypsinogen to trypsin, causing the activation of MMP-9. Trypsin is known to have a particular role in squamous cell carcinomas, in which MMPs are activated by trypsin-2, facilitating invasion and metastasis [73].”

After:

“When talking about DM and pancreatitis, two MMPs deserve special attention - MMP-2 and MMP-9 (gelatinases A and B, respectively), responsible for the degradation of gelatins, collagens III, IV, V, VII, X and XI, fibronectin, laminin, elastin, aggrecan, entactin and vitronectin [63]. In the course of acute and chronic pancreatitis, the expression of MMP-9 is increased [76] as a result of the intra-acinar conversion of trypsinogen to trypsin, causing the activation of MMP-9.”

Line 369 -Decellularization protocols typically involve a combination of  physical, chemical

You must reorganize the entire text between lines 369 and 444, linking the subjects in a fluid and logical way and harmonizing the discourse of various authors in a new text that is your text.

As in the first sentence, you begin your speech with the physical methods, you should exhaust the subject of physical methods and only then go on to enumerate other methods.

Line 369 -Decellularization protocols typically involve a combination of  physical, chemical

Ex: These two sentences address the same subject. You must integrate the two sentences into one.

Line 371 to 373 - followed by the separation of the ECM components through biological treatment, aiming at the solubilization of the cytoplasmic and nuclear components using detergents,

Line 375 to 377 - Decellularization by chemical methods, using acids or bases, occurs through the use of chemical agents that act in the solubilization of the cytoplasmic components of the cells and in the removal of nucleic acids such as DNA and RNA [77].

We appreciate the considerations, however, the paragraph corresponding to lines 369 to 374 corresponds to the total process of decellularization involving chemical, physical and biological methods. While the subsequent paragraphs describe each method (chemical (lines 375 to 407), physical (line 408 to 435), and biological (line 436 to 444) in isolation. Therefore, the authors see no need to change the text.

Line 389 to 390 - In the literature for decellularization of organs and tissues [99–101]. ??????????????

There was a typo and the sentence was changed.

Before:

“having several protocols mentioned. In the literature for decellularization of organs and tissues [99–101].”

After:

“having several protocols mentioned in the literature for decellularization of organs and tissues [102–104].”

  1. Tissue Recellularization

These cells can be obtained by biopsy from the patient's organ or donor organs [134]. However, the biopsy can generate a genetic defect, making them unviable and, ...

Could you please explain how biopsy cause a genetic defect?

The text has been revised and the excerpt has been removed from the text.

Before:

“These cells can be obtained by biopsy from the patient's organ or donor organs [134]. However, the biopsy can generate a genetic defect, making them unviable and, in the second case, there can be an immunological rejection of the donated organ. Furthermore, due to the low proliferative capacity of these cells, their application is limited [130].”

After:

“These cells can be obtained by biopsy from the patient's organ or donor organs [137]. However, due to the low proliferative capacity of these cells, their application is limited [133].”
